# Cluster-Assembled Nanoporous Super-Hydrophilic Smart Surfaces for On-Target Capturing and Processing of Biological Samples for Multi-Dimensional MALDI-MS

**DOI:** 10.3390/molecules27134237

**Published:** 2022-06-30

**Authors:** Emanuele Barborini, Giacomo Bertolini, Monica Epifanio, Alexander Yavorskyy, Simone Vinati, Marc Baumann

**Affiliations:** 1Nanotechnologies Unit, Materials Research and Technology (MRT) Department, Luxembourg Institute of Science and Technology (LIST), 5 Av. des Hauts-Fourneaux, 4362 Esch-sur-Alzette, Luxembourg; emanuele.barborini@list.lu; 2Tethis SpA, Via F.Olgiati, 20143 Milan, Italy; giacomo.bertolini@tethis-lab.com; 3National Centre for Sensor Research, Dublin City University, Glasnevin, D09 DXA0 Dublin, Ireland; monicaepifanio518@gmail.com (M.E.); alexander.yavorskyy@gmail.com (A.Y.); 4ParteQ GmbH, Brunnenstraße 12, 76316 Malsch, Germany; vinati@parteq.net; 5Meilahti Clinical Proteomics Unit, Institute of Biomedicine, Department of Biochemistry and Developmental Biology, Faculty of Medicine, University of Helsinki, Haartmaninkatu 2, 00014 Helsinki, Finland

**Keywords:** MALDI-MS, nanomaterials, reactive matrices, protein science

## Abstract

Matrix-assisted laser desorption/ionization mass spectrometry (MALDI-MS) on cluster-assembled super-hydrophilic nanoporous titania films deposited on hydrophobic conductive-polymer substrates feature a unique combination of surface properties that significantly improve the possibilities of capturing and processing biological samples before and during the MALDI-MS analysis without changing the selected sample target (multi-dimensional MALDI-MS). In contrast to pure hydrophobic surfaces, such films promote a remarkable biologically active film porosity at the nanoscale due to the soft assembling of ultrafine atomic clusters. This unique combination of nanoscale porosity and super-hydrophilicity provides room for effective sample capturing, while the hydrophilic-hydrophobic discontinuity at the border of the dot-patterned film acts as a wettability-driven containment for sample/reagent droplets. In the present work, we evaluate the performance of such advanced surface engineered reactive containments for their benefit in protein sample processing and characterization. We shortly discuss the advantages resulting from the introduction of the described chips in the MALDI-MS workflow in the healthcare/clinical context and in MALDI-MS bioimaging (MALDI-MSI).

## 1. Introduction

Matrix-assisted laser desorption/ionization mass spectrometry (MALDI-MS) is one of the most desired analytical tools both in research and clinical diagnostics [1] to detect and analyze biomolecules of any kind, such as peptides, proteins, nucleic acids, and carbohydrates, as well as cells, viruses, bacteria, fungi, and whole histological sections [2], due to its high ionization efficiency, sufficiently high sensitivity, and most of all, its rapidity [3,4]. Today, a modern MALDI instrument can deliver easily up to 10,000 laser shots per second, while around 10,000–20,000 shots per second is enough for one analyte/sample to be detected, i.e., the speed of one sample per a second cannot be beaten by any other mass spectrometer of today. Moreover, the cost for running a MALDI-MS instrument is basically only based on the cost of the matrix chemical, sample plates, and service. Interestingly, in recent years MALDI-MS applications are spreading far from their original protein chemical, proteomic, and molecular biology applications, towards healthcare and clinical areas such as pathogen analysis [2], where the rapidity in the identification of species responsible of infections may greatly help to monitor and select the use of antibiotics for severe clinical conditions of bacteriaemia. Nonetheless, MALDI-MS analysis efficiency depends on various conditions such as the complexity of the sample, the presence of contaminants, and the quality of the crystallization and homogeneity of the matrix–analyte mixtures. Non-optimal conditions could lead to a significant decrease in the sensitivity and reproducibility of the analysis.

In comprehensive and deep protein analyses, the commonly adopted protocol for sample preparation requires a long workflow that includes protein extraction, denaturation, reduction, alkylation, cleaning, and enzymatic cleavage. In addition, HPLC or UHPLC is frequently used for final pre-fractioning of complex samples prior to MALDI-MS analysis. However, HPLC (UHPLC) is time consuming and may suffer also from other drawbacks, such as unexpected contamination of the analytes and not being able to detect a sufficiently high coverage of the digested peptides for a proper protein identification. In “shotgun proteomics”, the cleavage of proteins into smaller fragments by highly specific enzymes, such as trypsin, is a crucial step for successful protein identification. Protein digestion is usually carried out in solution, in selected test tubes overnight. Much effort has been put into the development of rapid and more accurate digestion systems [5]; however, a well-defined approach has not yet been agreed upon, partially attributed to the poor stability of trypsin throughout the various sample treatment steps. Additionally, the loss of a large number of released peptides in the sample digestion process (e.g., by capture on vial plastic/glass walls) often hampers the desired percentage of peptide coverage. Trypsin digestion columns are available from different companies, but their success has been limited by their rather short lifetime due to trypsin stability in the column [6,7].

Considerable efforts have been devoted to the design and fabrication of nanostructured layers with functional properties by regulating and controlling the surface chemistry, resulting in very promising substrates for protein microarray applications and third-generation point-of-care (POC) biosensors [8,9,10,11,12]. In particular, nanostructured metal oxides allow for the fine tuning of physical and chemical properties, such as surface wettability and amphiphilicity, which have a remarkable role in regulating the interaction between the nanostructured surface and the biological matter [13,14,15,16,17,18,19,20]. A growing number of researchers have paid attention to TiO2-derived materials with superwettability [21]. Clearly, multifunctional TiO2 materials is an inevitable trend. For instance, Gailite et al. [20] demonstrated that proteins such as trypsin can stably be bound on nanostructured films by physical absorption, i.e., without the need of any chemical binder, while still maintaining their activity. This is based on the fact that the features of the nanostructured surface strongly influence both the binding strength at the protein–surface interface and the amount of protein absorbed [20,22,23,24]. Moreover, adjusting the surface properties is crucial to create various biobinders, such as cell interacting surfaces [25]. These results suggest that the morphological features of nanostructured metal oxides layers can be deemed as a preferable environment, not affecting the original biomolecule features, where biomaterial can be reliably stored and processed.

Here, we report a study aiming to develop innovative MALDI-MS targets, where the process of soft-assembling of ultrafine TiO_2_ clusters is exploited to produce an array-patterned functional layer characterized by a nano-porous structure in which the effective capture of biological samples can take place (biobinders). Supersonic cluster beam deposition (SCBD) was adopted for the direct integration of the cluster-assembled film on suitable polymeric substrates. The phenomenon of UV photo-induced super-hydrophilicity occurring in Group IV metal oxides [26] was exploited to generate a large hydrophilic-hydrophobic discontinuity of surface wettability at the border of the TiO_2_ areas, which acts as a containment structure for aqueous droplets of samples/reagents. This enables easy and reliable on-chip sample processing, either manually or through automatic liquid handling, before and during MALDI-MS analysis. The nanoscale surface roughness of the films assembled by ultrafine clusters were determined out to be a key morphological feature to control the crystallization phenomenon occurring in MALDI matrix deposition, with unexpected advantages in terms of area coverage uniformity, reproducibility, and homogeneity of crystals sizes. Such features could also be especially important for the increasing usage of MALDI instruments for automated MALDI imaging analyses (MSI), increasing their spatial resolution and homogeneity. The chemical inertness of TiO_2_ makes it compatible with basically any kind of chemical treatment, including those precluded from generally used MALDI plates, such as many organic solvents.

## 2. Material and Methods

### 2.1. Samples, Reagents and Procedures

The biological samples used throughout this research were human serum albumin (HSA, Boehringer Diagnostics, ORHA 20/21 grade), bovine serum albumin (BSA, Sigma-Aldrich/Merck, Burlington, MA, USA, >99%), MS standard grade trypsin digest of bovine serum albumin (polypeptide mixture standard, Bruker, Billerica, MA, USA), and human gelsolin samples from patient cell cultures of gelsolin amyloidosis (AGel) patients. Digestions were carried out with modified trypsin (Promega, Madison, WI, USA, V5111, sequencing grade) in ammonium bicarbonate buffer (Fluka, Buchs, Switzerland, 99.5%) at concentrations in the range 10–40 mM, with 5% acetonitrile (ACN; Fisher Chemical, Hampton, NH, USA, 99.9%). The enzyme-to-protein mass weight ratio was set up in the range of 1:20–1:50. The MALDI matrices used were sinapic acid (SA, Sigma, St. Louis, MO, USA, >99%) for proteins and α-Cyano-4-hydroxycinnamic acid (CHCA, Sigma, >99%) for peptides; both were prepared as saturated solutions in 1% TFA and ACN in 2:1 volume ratios. MALDI matrices were prepared freshly immediately before each MALDI-MS analysis session and used for no longer than 24 h after preparation.

### 2.2. Array-Patterned Ns-TiO_2_ Chips

Chips were produced at Tethis S.p.A. production facilities in Milan, Italy, by depositing cluster-assembled nanostructured TiO_2_ (ns-TiO_2_) on conductive carbon-filled polypropylene (PP) substrates having a volume resistivity of about 10^2^ Ω·cm, dimensions of 75.0 × 25.0 ± 0.5 (mm), and a thickness of 1.0 mm (Stratec Consumables, Anif, Austria). Conductive substrates are required by some frequently used MALDI-MS instruments based on time-of-flight (TOF) ion analyzers (e.g., Bruker UltrafleXtreme, Bruker RapifleX, or Shimadzu AXIMA Performance), nevertheless they can also be used in instruments equipped with different ion analyzers (e.g., Thermo Orbitrap (Waltham, MA, USA) and Waters Synapt (Milford, MA, USA) Series MS with MALDI Source).

Ns-TiO_2_ films were deposited by supersonic cluster beam deposition (SCBD), using a pulsed micro-plasma cluster source (PMCS) [27,28,29]. This method offers the remarkable advantage of allowing the direct integration of ultrafine oxide clusters on functional areas of devices in one step, at room temperature, avoiding the use of any chemical ligand as well as any post-deposition processing (e.g., the high temperature calcination step of sol-gel methods or curing step of xerogel methods). PMCS exploits the gas-dynamic confinement of a high-voltage, high-current, pulsed electric discharge to vaporize titanium atoms from a metallic rod (Goodfellow, Livermore, CA, USA, 99.6%). Thermalization and subsequent aggregation of Ti atoms in clusters occur in a pure argon atmosphere (Linde Gas, Dublin, Ireland, 6.0). Finally, the mixed argon clusters undergo a pressure-driven supersonic expansion through PMCS nozzle, towards a vacuum chamber where it forms a collimated cluster beam.

By exposing the polymeric substrates to the cluster beam, the growth of a cluster-assembled nanostructured film takes place. Thanks to beam collimation, the film can be easily patterned by the stencil mask approach with micrometric lateral resolution [30]. The kinetic energy of the clusters in the beam is sufficiently low to prevent fragmentation as well as surface diffusion, ensuring the growth of the film to occur according to the so-called ballistic model [31]. In this respect, the film shows a hierarchical porous structure, where pore sizes range from the nanometer up to hundreds of nanometers [32]. The morphological features of ns-TiO_2_ film are visible in Figure 1 (top inset). A thickness of about 200 nm was chosen for this study. Through the use of stencil mask patterning, nanomaterial was deposited in form of an array of dots having diameter of 2 mm and spacing of 4.5 mm, according to the multi-well plates standard (Figure 1). An additional set of dots with a diameter of 1 mm positioned within the main array was also included. This second set of dots was intended to provide distributed points throughout the whole device surface for calibration standards.

Before use, chips were irradiated with UV (λ = 254 nm) in a UV lamp equipped fume hood in order to induce super-hydrophilicity in ns-TiO_2_. Given a lamp power of 30 W and a lamp–chip distance of about 40 cm, the irradiation time to activate the super-hydrophilic state is less than half an hour.

### 2.3. In-Vial and On-Chip Comparison

The experiments aiming to directly compare in-vial and on-chip digestions were carried out as follows: The digestion solution was prepared in a vial with 45 µL of ammonium bicarbonate 11.1 mM, 2.5 µL of HSA at a concentration of 2.64 µg/µL, and 2.5 µL of trypsin at a concentration of 0.05 µg/µL. Immediately after the enzyme was added to the digestion solution, 5 µL was taken out of the vial and pipetted on the chip. Sample digestion then proceeded in parallel in-vial and on-chip for 2 h at 40 °C. To limit buffer evaporation on-chip, we provided a high humidity environment with a water thermal bath just below the chip.

### 2.4. Multi-Dimensional MALDI

Multi-dimensional MALDI experiments were carried out as follows: 2 µL of sample in water solutions were pipetted on ns-TiO_2_ dots and let dry (concentration of 1 µg/µL for BSA and Ovalbumin. Then, 1.5 µL of the SA solution was added and let dry. After MALDI-MS data collection on the protein mass range, the SA matrix was removed/washed from the dots through a washing step with two repetitions, where 2 µL of methanol 70% was pipetted, moved back and forth 10 times with the pipette plunger, and finally discarded. A total of 5 µL of ammonium bicarbonate 40 mM was then pipetted on the dots and 1 µL of trypsin solution was added. The chip was subsequently incubated in a closed box (to avoid droplet drying) at 45–50 °C for 30 min and let dry at room temperature. Finally, 1.5 µL of CHCA solution was then added and let dry.

For the verification of possible disulfide bridges in the AGel sample, the tryptic peptide digested sample was washed as described above and 10 µL of 100 mM DTT added to the sample dot. Samples were incubated with DTT for 10 min at 60 °C after which time the DTT solution was removed by pipetting. A total of 1.5 µL of CHCA solution was then added and let dry.

### 2.5. Identification of Amino Acid Sequences in Poorly Soluble Amyloidogenic Fragments

The experiment on the identification of the amino acid sequences in poorly soluble amyloidogenic fragments in the clinical samples was carried out as follows: Human AGel cells were cultivated as described elsewhere [33]. A 3 cm diameter plate was used for the cell culturing. At full confluence, cells were harvested by mechanical scratching and subsequently moved to sample vials. The cells were washed three times in ice-cold physiological phosphate-buffered saline (PBS) buffer, after which the cells were treated according to the instructions of the manufacturer for immunoprecipitation (IP) (Abcam laboratories). In brief, monoclonal anti-human gelsolin antibodies were bound to magnetic beads (Thermo Scientific, Waltham, MA, USA) according to the manufacturer’s instructions and then used for the IP. Bound gelsolin was eluted from the beads by pH 2 glycine-buffer shock, quickly neutralized and then acetone precipitated at −20 °C for two hours.

Precipitated protein was applied to the chip and digested with trypsin, as described above. As a control, the same amount of bovine gelsolin was applied to a regular MALDI-MS stainless steel sample plate (bovine and human gelsolin share amino acid identity on the sequence area of patient mutation).

### 2.6. Mass Spectrometry

MALDI-MS analysis was carried out with a Bruker UltrafleXtreme 2000 Hz mass spectrometer operating in reflector mode. Standard stainless-steel slide adapters were used as sample holders. Mass spectra were acquired as a sum of 5000–10,000 laser shots operating at a frequency of 2000 Hz in partial random sampling mode and peak intensities in the range of 10^5^ counts were regularly achieved. The internal calibration method was adopted in experiments involving well-characterized samples (i.e., HSA, BSA, and BSA trypsin digest), while in other cases, MS calibration standard mixtures (Bruker Daltonics GmbH & Co, Bremen, Germany) were added to the chips. The instrument was operated by Flex-Control version 3.4, spectra analysis was performed by Flex-Analysis version 3.4 (Bruker Daltonics GmbH & Co, Bremen, Germany), and data from spectra analysis were processed for protein identification in Bio-Tools version 3.2 (Bruker Daltonics GmbH & Co, Bremen, Germany).

## 3. Results and Discussion

### 3.1. Hydrophilic-Hydrophobic Containment Structures

UV irradiation generates a photo-induced super-hydrophilic state in TiO_2_, according to a complex surface phenomenon where photo-generated hydroxyl (OH) groups from environmental water determine nanoscale structural changes on material surface, which in turn cause the complete spreading of water droplets, as a sort of two-dimensional capillary phenomenon [13,26]. At the same time, UV irradiation does not cause either the modification of the wettability of polymeric substrates, which maintain their original highly hydrophobic character, nor the release of volatile compounds from the polymer, which may affect the super-hydrophilic feature of ns-TiO_2_. UV treatment hence resulted in super-hydrophilic regions embedded in a hydrophobic surrounding. Super-hydrophilic dots literally capture the droplets from the pipette tip and ensure their uniform distribution on dot areas, while the border between the hydrophilic dots and the hydrophobic surrounding provides a barrier acting as an effective containment structure for droplets, as shown in Figure 1 (bottom inset). It was determined that the structuring of the chip surface in hydrophilic nano-porous dots and hydrophobic surroundings allows the effective and reliable capture and absorption of samples on well-defined chip positions as well as the possibility of on-chip processing of the captured samples, i.e., such dots act as sample reactors.

The stability of the UV-activated super-hydrophilic state in ns-TiO_2_ has been demonstrated to last for several days, a time much longer than that observed in microcrystalline TiO_2_ where the recovery to a more hydrophobic state occurs in few hours [13]. We argue that this is a direct consequence of the cluster-assembled film structure, whose grain composition can effectively accommodate the mechanical stresses of structural modification induced in the material by UV irradiation that are responsible of the spontaneous recovery from the metastable super-hydrophilic state to a more hydrophobic state.

### 3.2. Nanostructure Driven Matrix Crystallization

Matrix crystallization is generally known to be a critical step in the preparation of samples for MALDI-MS analysis. Big clumps of crystals irregularly distributed across the laser shooting area create unfavorable conditions for repeatable MS data collection. Vice versa, finely grained, homogeneous, and uniformly distributed crystals on well-defined and repeatable areas create the ideal situation. We observed that the cluster-assembled surface is able to drive matrix crystallization phenomena towards an optimal scenario by favoring the formation of smaller, more homogeneous in size, and uniformly-distributed crystals with respect to non-nanostructured surfaces, as shown in Figure 2. This may be mostly related to surface roughness at the nanoscale, which offers a huge number of “point-like surface defects” where crystals first nucleation can begin.

### 3.3. In-Vial vs. On-Chip Protein Digestion

Containment effect was exploited by carrying out trypsin digestion of HSA (human serum albumin) on-chip. We performed the same digestion step (2 h at 40 °C) in a standard vial in order to compare the peptide distribution resulting from the two processes. Actually, it is well known that the digestion of proteins in a vial may suffer the loss of those peptides that have a strong affinity to hydrophobic plastic surfaces. On the contrary, a digestion taking place on the same chip where MALDI-MS analysis will be successively performed, has the obvious advantage of avoiding any fraction loss. Moreover, since it was demonstrated that ns-TiO_2_ is able to effectively absorb and concentrate proteins [14,16,20], we expected that at the end of digestion that all peptides generated would be effectively absorbed and concentrated in the ns-TiO_2_ dots. In this respect, ns-TiO_2_ dots can be regarded as a sort of active “solid-state micro-wells”.

Figure 3 shows the peptide mass spectra of on-chip digested and in-vial digested HSA, where the former appears richer in peaks than the latter. To quantitatively highlight this observation, we listed the 18 most relevant peaks belonging to both spectra and we normalized their intensities to the intensity of the maximum, which is the peak at 1899 Da in both spectra (Figure 4). Apart from a group of four peaks, whose normalized intensities are roughly the same after the two digestions (arrows), all other masses systematically show a much lower normalized intensity in the spectrum of in-vial digestion. The logarithmic scale of the vertical axis highlights the huge decrease in normalized intensities. This result can be explained by the tendency of several peptides to be lost during in-vial processing and emphasizes one of the benefits of on-chip processing. The trend of losing peptides during in-vial processing was confirmed by analyzing in-vial digests after 5 days: several peptides had vanished either totally or to a great extent while on-chip samples stayed quasi-stable (data not shown).

### 3.4. Multi-Dimensional MALDI

Multi-dimensional MALDI is a recently coined term that refers to the capability to collect MS information from a biological sample at the level of the intact proteins as well as at the level of the peptide ensemble generated by the enzymatic cleavage of the proteins of exactly the same sample. A multi-dimensional MALDI analysis could also include a peptide fragment pattern with first a non-cleaved disulfide bond and then with the same peptides released from their disulfide bonds. Moreover, the concept of multi-dimensional MALDI can be applied, for instance, to complex bacteria samples whose MS profile is not clear enough to identify the involved species; the collection of additional MS information after enzyme digest of exactly the same sample may help solve this issue. The motivation for performing multi-dimensional MALDI is the possibility to effectively remove (wash out) the MALDI matrix after “protein level” MS data collection, while at the same time ensuring that the original sample is not eluted. Matrix removal must achieve the condition where acid residuals would not affect the basicity of the trypsin buffer of the subsequent digestion step. We observed that such a condition is fulfilled by ns-TiO_2_ due to a combined action of the nano-porosity and the affinity of the material to biomolecules and bio-entities in general. The multi-dimensional MALDI approach was successfully applied to BSA. Results on BSA are shown in Figure 5.

### 3.5. Gelsolin Amyloid Peptide Detection

Hereditary gelsolin amyloidosis is an autosomal dominantly inherited systemic disease, first described in 1969 by the Finnish ophthalmologist Jouko Meretoja. The estimated number of disease carriers in Finland is close to 1000 patients. Although originally detected in Finland, it is now increasingly reported in other countries in Europe, North and South America, and Asia. AGel is caused in most kindred by a c. 640G>A point mutation in the gene coding for gelsolin (GSN). Comparable to other amyloidogenic protein folding disorders such as Alzheimer’s disease, Parkinson’s disease, and Huntington’s disease, the formation of oligomeric aggregates of a protein fragment of gelsolin during the amyloidogenesis process with proteotoxic and cytotoxic effects has been suggested to have pathogenetic significance in AGel amyloidosis [34,35,36]. Such amyloidogenic mutations usually occur in sequence areas with low solubility under physiological conditions, making their detection and research difficult. Staining with Congo red dye (CR) is one of the major methods used to detect the amyloid structure of protein aggregates. However, a series of experiments have shown that CR staining is insufficient for confirmation of the amyloid nature of protein aggregates. Mass spectrometry on Alzheimer’s amyloid a-beta fragment, for example, is usually performed by measuring a soluble fragment of the enzymatically digested a-beta peptide fragment, not the intact amyloid itself [37]. If digested by enzymes, they tend to “vanish” from the solution due to their strong character to bind to surfaces with nucleation potential, such as test tube walls. Therefore, an enzymatic digest in a regular test tube will cause an unbalanced and partially undetectable signal of these amyloidogenic peptide fragments in any mass spectrometric analysis. Additionally, commonly used reverse-phase column chromatography experiments usually result in no or very low detection of most amyloidogenic peptide fragments.

Altogether, we purified approximately 200 ng of human gelsolin by immuno-precipitation (IP) from human cell cultures. This was enough to perform an on-chip digest and to measure and identify the peptide with the mutation sequence. As shown in Figure 6, a tryptic peptide with the *m*/*z* of 3922.961 Da could be detected in the on-chip performed digest (Figure 6B) but not in the corresponding sample digest prepared in a standard test tube and pipetted on the standard stainless steel sample plate (Figure 6A). The measured mass and subsequent MS-MS analysis identified the sequence ATEVPVSWESFNNGDCFILDLGNDIYQWCGSSSNR, which includes the mutation site of D/N 187. This result clearly indicates that hydrophobic peptides, usually tempted to attach onto test tube surfaces, such as often the amyloid core peptides, can quite easily be detected if the digestion is carried out on-chip.

### 3.6. Multi-Dimensional Detection of Disulfide Bridges

We used the on-chip sample processing approach to further investigate whether gelsolin would have a disulfide bond between the C (188) and any second free cysteine in the sequence. For this, we first performed a tryptic digest on the chip without any reducing agent, recorded the peptide map under non-reducing conditions and then washed out the first peptide matrix. Then, we added dithiothreitol (DTT) to the sample, incubated it further for 10 min at 60 °C, added a new layer of peptide matrix, and measured the very same sample again. As a result of this, we would have had the possibility to observe two peptides to separate if any disulfide bond would have existed which DTT would have broken in the second treatment phase. No such peptide shifts were observed. With such a simple experiment we could beautifully show that indeed, there is no disulfide bond to the free cysteine in position 188.

## 4. Conclusions

In the present work, super-hydrophilicity was induced in nanoporous TiO_2_ by UV irradiation in order to setup a hydrophilic-hydrophobic barrier at the dots’ border, which provides effective and highly reliable confinement of sample/reagents droplets, as well as uniform distribution of sample compounds on dot area upon drying. Our results show that material nanoporosity and bioaffinity cooperate for the purpose of sample capture while nanoscale surface roughness drives matrix crystallization process to generate fine, homogeneous, and uniformly distributed crystals. This extraordinary feature of interactive binding and matrix crystallization can be of paramount importance for, e.g., pathogen analysis by MALDI-MS, where automated sample reading is often based on time restriction and a “no signal” may give a false negative result due to the instrument not finding the target analyte hotspot in the given time. Moreover, automated sample pre-cleaning is often desired for unwanted contaminant removal. Nanoporosity and bioaffinity are also of special interest for whole-tissue sample MALDI imaging mass spectrometry (MSI) where it is important to have strong and homogenous binding of the biomaterial to the chip surface and homogenous matrix distribution for optimal spatial resolution. Improved adhesion avoids detachment, tearing, and shrinking during the tissue processing. Small molecular pharmaco-kinetic studies with MSI, where any kind of tissue treatment should be excluded, is one more obvious target for our advanced surface engineered reactive containments [38].

In addition to the advantage offered for trypsin on-chip digestion, where peptide fractions are not lost due to hydrophobic capture on vial walls, the possibility for repeated sequences of sample processing steps is a major advantage of the multi-dimensional MALDI-MS platform. In our test setup, we have extended our experiments only to the level of second-to-third dimensional MALDI-MS, but there are no restrictions to use our chip material for further subsequent reactions.

In addition to proteins, the hierarchical structure of the porosity of cluster-assembled TiO_2_, ranging from few nanometers up to a hundred of nanometers, can be efficiently exploited for the capture of bioentities larger than biomolecules, such as exosomes, extracellular vesicles (EVs), viruses, bacteria, and various cell types. As in the case of present pathogen analysis technology on MALDI-MS, the collection of information from such complex samples can be managed with non-enzymatic MS profiling and extended database comparison. In the case of pathogen analysis, multi-dimensional MALDI-MS could be used for automated pathogen subtyping.

Combining these new MALDI-MS chips with a liquid-handling robotic system will simplify the detection of biomarkers for MS profiling of liquid biopsy-based biological samples. Our results may reveal unprecedented opportunities for MALDI-MS analysis in healthcare areas such as microbiology, hematology, and general liquid biopsy-based oncology.

## Figures and Tables

**Figure 1 molecules-27-04237-f001:**
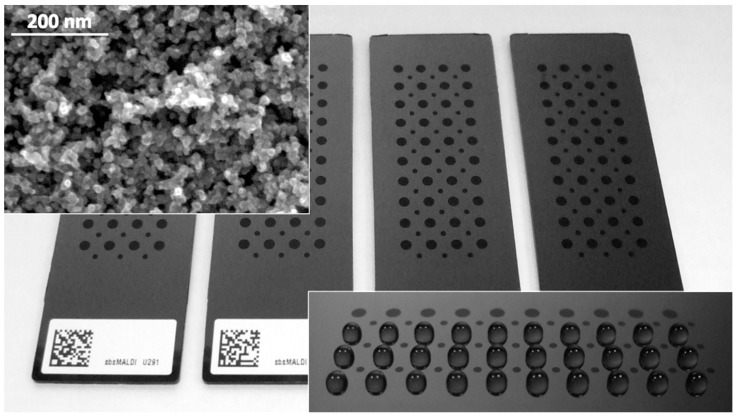
Picture of the dot-patterned slides used in this study (the substrate shown here is carbon-filled polypropylene). Major dots have a diameter of 2 mm and spacing of 4.5 mm, while minor dots have a diameter of 1 mm and the same 4.5 mm spacing. Nanomaterial thickness is about 200 nm. Top inset shows a scanning electron microscopy (SEM) image of ns-TiO_2_ surface. Bottom inset shows the confinement of aqueous droplets due to the hydrophilic-hydrophobic barrier, which acts as effective and well-defined containment structure (as a sort of “solid-state micro-well”) for sample capture and on-chip processing.

**Figure 2 molecules-27-04237-f002:**
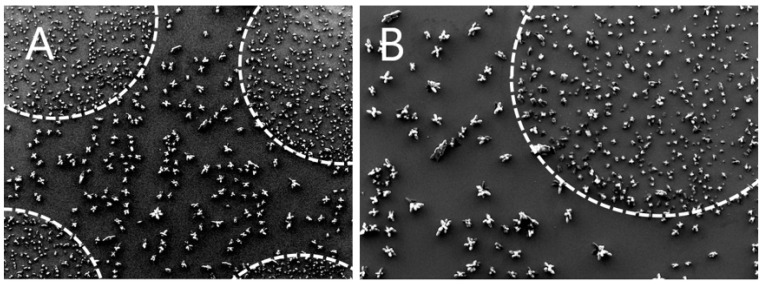
Scanning electron microscopy (SEM) images of CHCA crystals. Image (**B**) is a detail of the upper right corner of the left image (**A**). Pictures allow for direct comparison of matrix crystallization on areas having nanoscale surface roughness (dashed areas) and on surrounding flat surface.

**Figure 3 molecules-27-04237-f003:**
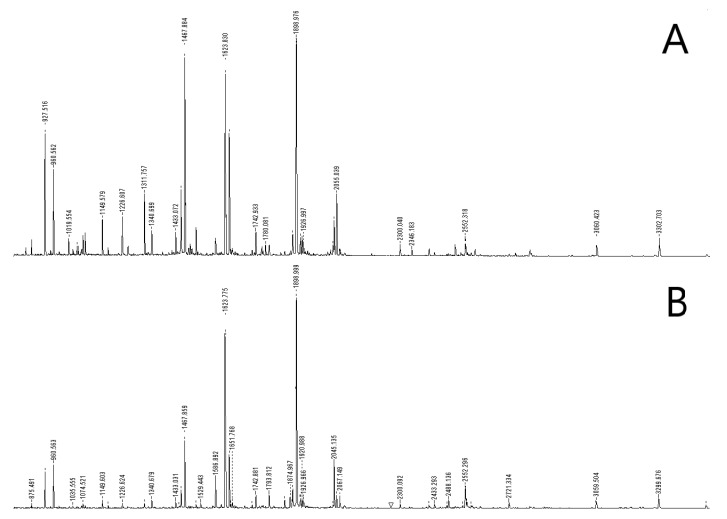
HSA peptide mass spectra after on-chip trypsin digestion (**A**) and in-vial digestion (**B**).

**Figure 4 molecules-27-04237-f004:**
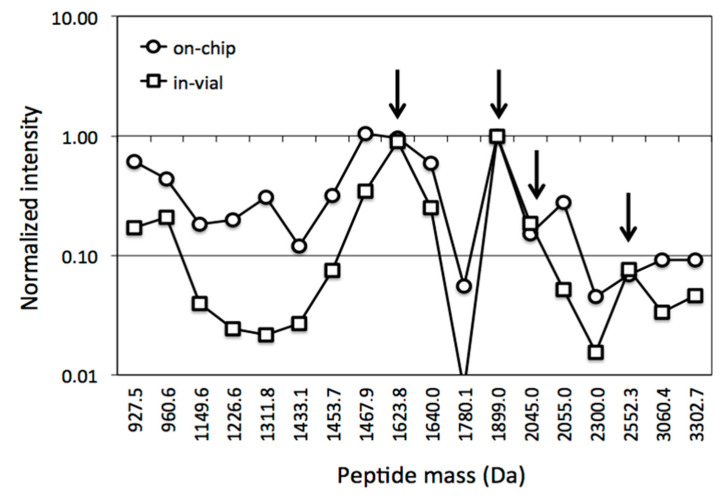
Intensities of the 18 main peaks belonging to both spectra in Figure 3 after normalization with respect to the intensity of the maximum one (at *m*/*z* of 1899 Da). Lines connecting data points have no experimental meaning: they only aim to guide the eye. Apart from a group of four peaks, whose normalized intensities are the same in the two digestions (arrows), i.e., they are not affected by the interaction with the vial walls, all others systematically show a lower normalized intensity in the spectrum related to in-vial digestion. Logarithmic vertical axis helps to show smaller normalized intensities and highlights the huge overall decrease in peptide intensities from in-vial processing.

**Figure 5 molecules-27-04237-f005:**
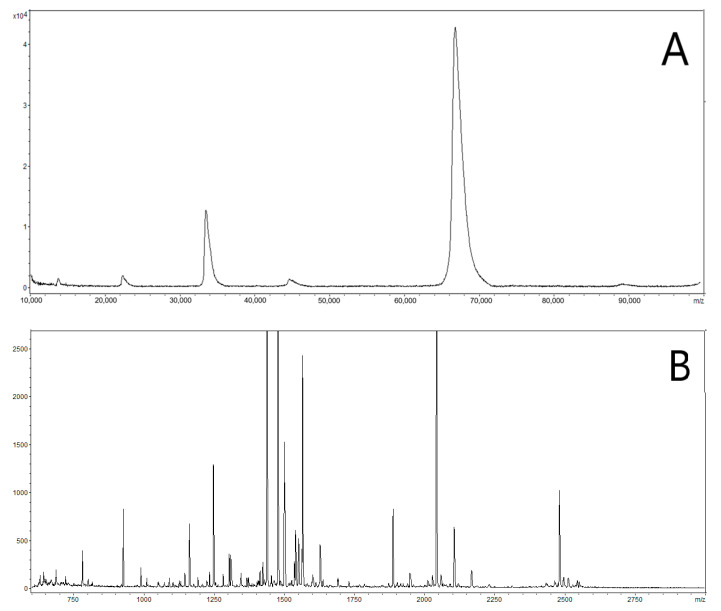
Mass spectra of untreated BSA (**A**) and BSA tryptic digest (**B**). The latter was collected on the same spot of the former, after the removal of SA matrix with a 70% methanol solution and on-chip digestion at 45–50 °C for 30 min.

**Figure 6 molecules-27-04237-f006:**
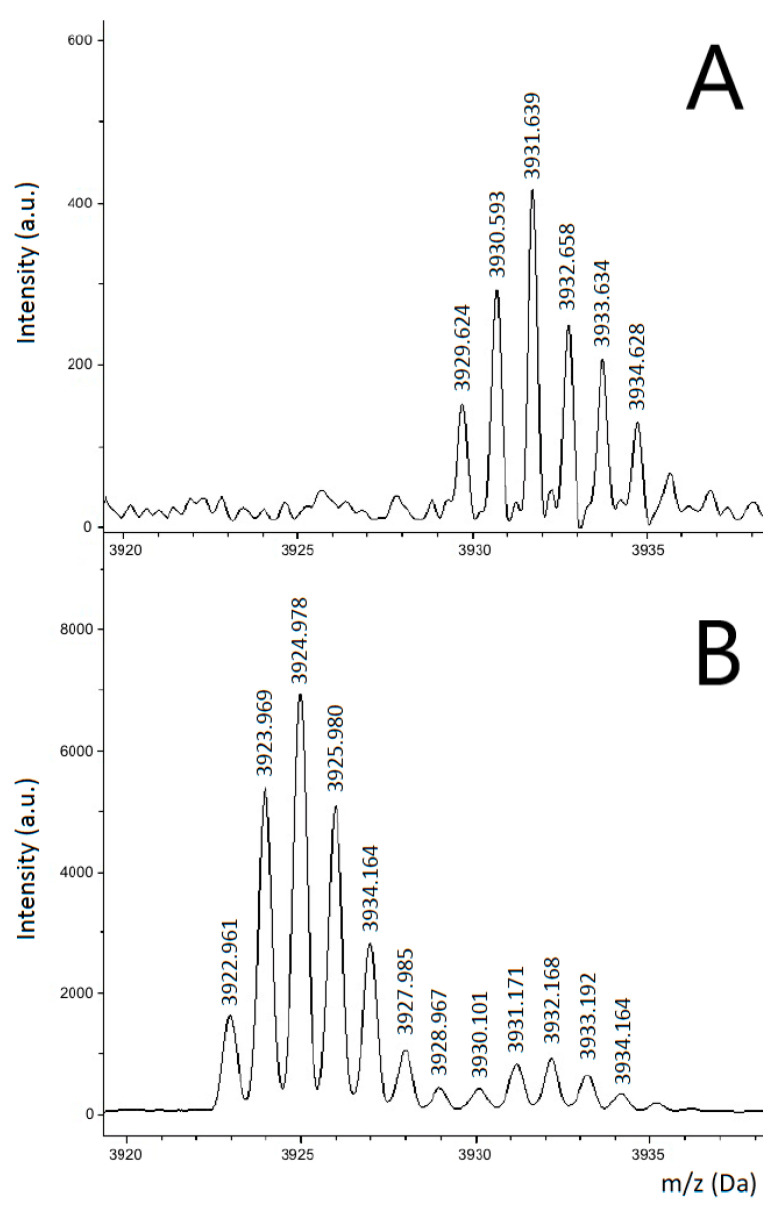
MALDI mass spectra of gelsolin tryptic digest according to in-tube standard procedure (**A**) in comparison to the digest of the same sample carried out on-chip (**B**). The peptide peak structure at *m*/*z* of 3921.267 Da is visible in the on-chip digested sample, while it cannot be detected in the in-vial digested sample. This peak structure corresponds to the sequence ATEVPVSWESFNNG**D**CFILDLGNDIYQWCGSSSNR, which includes the mutation site of D/N 187.

## Data Availability

Not applicable.

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
