# Peer review of "Cluster-Assembled Nanoporous Super-Hydrophilic Smart Surfaces for On-Target Capturing and Processing of Biological Samples for Multi-Dimensional MALDI-MS"

_molecules, 2022, doi:10.3390/molecules27134237_

Round 1

Reviewer 1 Report

This work reported a study aiming to the development of innovative MALDI-MS targets, where the process of soft-assembling of ultrafine TiO2 clusters was exploited to produce an array-patterned functional layer, characterized by a nanoporous structure in which the effective capture of biological samples can take place (Biobinders). It is well planned, the results are properly described and discussed. Overall, I recommend this manuscript for publication in Molecules after some major revisions towards the following points:

1. In the abstract and conclusion sections, the author needs further condense and summarize.

2. In the introduction section, the author should introduce some new related literatures on nanostructured metal oxides that can tune physical and chemical properties such as surface wettability and amphiphilicity.

3. Overall discussion in the nanostructure driven matrix crystallization and gelsolin amyloid peptide detection is ok however it should be improved, especially the work mechanism of gelsolin amyloid peptide detection.

4. The novelty of this work should be further clearly presented.

5. There are still some problems to be corrected, such as “results and discussions” should be “results and discussion”, the format of references, et al.

6. For the description of the figures, I recommend replacing the top/bottom of the figure with figure A/B.

Author Response

Dear Reviewer 1: We would like to thank you for your valuable comments and adds to our manuscript. We have now tried to address all the comments and criticism as much as possible. In detail:

  1. In the abstract and conclusion sections, the author needs further condense and summarize. >> We have condensed both the abstract and the conclusion sections according to this suggestion (indicated changes in red and in the tracking section in the resubmitted version).
  2. In the introduction section, the author should introduce some new related literatures on nanostructured metal oxides that can tune physical and chemical properties such as surface wettability and amphiphilicity. >> We have added two new references 21 and 25 and also added the corresponding text into the manuscript (indicated changes in red and in the tracking section in the resubmitted version).
  3.   Overall discussion in the nanostructure driven matrix crystallization and gelsolin amyloid peptide detection is ok however it should be improved, especially the work mechanism of gelsolin amyloid peptide detection. >> We have tried to improve the text accordingly and added new text to explain in more detail the work mechanism of amyloid peptide detection. We also added a new reference to this with the number 37 (indicated changes in red and in the tracking section in the resubmitted version).
  4.   The novelty of this work should be further clearly presented. >> We have added more text on the novelty of this work in the new conclusion section. (All indicated changes in red and in the tracking section in the resubmitted version).
  5.  There are still some problems to be corrected, such as “results and discussions” should be “results and discussion”, the format of references, et al. >> We have corrected these problems.
  6. For the description of the figures, I recommend replacing the top/bottom of the figure with figure A/B. >> We have added letters to all relevant figures and removed the top/bottom texts.

We hope that with these changes the reviewer 1 comments are properly addressed. However, we are more than happy still to add or remove any text, if needed. e.g., we are experts on various amyloidoses and thus have much more data to add to this topic, if necessary.

Reviewer 2 Report

  1. The authors have demonstrated a novel approach for on-board sample processing in MALDI-MS analysis by integrating dot-patterned arrays of TiO2 nanostructure. With this approach, they successfully showed that on-chip trypsin digestion can preserve more peptide fractions compare with in-vial digestion. They also carried out multi-dimensional MALDI for untreated BSA and then BSA tryptic digest. Overall, this is a well written manuscript in a contemporary topic, I recommend publication after addressing the following comments.
  2. the authors already told a very clear story with nice figures, detailed procedures and thorough discussions. One small thing to add is that in 3.6, the authors used the on-chip sample processing for multi-dimensional detection of disulfide bridges, the result of the MALDI spectra before and after DTT should be presented in a figure.
  3. Line 73, These papers should be cited when talking about the efforts devoted to the design and fabrication of nano-structured layers with functional properties: (1) J. Am. Chem. Soc. 2018, 140 (7), 2421–2425. https://doi.org/10.1021/jacs.7b13245; (2) Lee, D., Kim, Y., Jalaludin, I., Nguyen, H. Q., Kim, M., Seo, J., ... & Kim, J. (2021). MALDI-MS analysis of disaccharide isomers using graphene oxide as MALDI matrix. Food Chemistry342, 128356; (3)  Analyst 2019, 144 (21), 6321–6326. https://doi.org/10.1039/c9an01113g.

Author Response

Dear reviewer 2: We would like to thank you for your valuable comments and adds to our manuscript. We have now tried to address all the comments and criticism as much as possible. In detail:

  1. the authors already told a very clear story with nice figures, detailed procedures and thorough discussions. One small thing to add is that in 3.6, the authors used the on-chip sample processing for multi-dimensional detection of disulfide bridges, the result of the MALDI spectra before and after DTT should be presented in a figure. >> We indeed used the chip for to detect the possible disulphide bridge in gelsolin amyloid peptide position 188, but since the result was negative (the free cysteine was not bound to another peptide) we did unfortunately not record this in our data file. It would just have been two identical figures. We have however, added a new text about the method of verifying a disulphide bridge, into the Material and Methods section in the 2.4 Multi-dimensional MALDI paragraph. We also added a few words to the Results and Discussion section about this.
  2.   
    1. Line 73, These papers should be cited when talking about the efforts devoted to the design and fabrication of nano-structured layers with functional properties: (1) J. Am. Chem. Soc.2018, 140 (7), 2421–2425. https://doi.org/10.1021/jacs.7b13245; (2) Lee, D., Kim, Y., Jalaludin, I., Nguyen, H. Q., Kim, M., Seo, J., ... & Kim, J. (2021). MALDI-MS analysis of disaccharide isomers using graphene oxide as MALDI matrix. Food Chemistry342, 128356; (3)  Analyst 2019, 144 (21), 6321–6326. https://doi.org/10.1039/c9an01113g. >> We have added both references to the manuscript with the numbers 8 and 9. We actually already originally thought to add these two references to our manuscript but felt then that they were a little more specific to reflect a general view of our work. Nevertheless, we fully agree on this add.

Round 2

Reviewer 1 Report

I recommend the revised manuscript for publication in Molecules.